# COVID-19 and the Gaping Wounds of South Africa’s Suboptimal Immunisation Coverage: An Implementation Research Imperative for Assessing and Addressing Missed Opportunities for Vaccination

**DOI:** 10.3390/vaccines9070691

**Published:** 2021-06-23

**Authors:** Chukwudi A. Nnaji, Charles S. Wiysonge, Maia Lesosky, Hassan Mahomed, Duduzile Ndwandwe

**Affiliations:** 1Division of Epidemiology and Biostatistics, School of Public Health and Family Medicine, University of Cape Town, Anzio Road, Observatory, Cape Town 7925, South Africa; charles.wiysonge@mrc.ac.za (C.S.W.); maia.lesosky@uct.ac.za (M.L.); 2Cochrane South Africa, South African Medical Research Council, Francie Van Zijl Drive, Parow Valley, Cape Town 7501, South Africa; duduzile.ndwandwe@mrc.ac.za; 3Division of Epidemiology and Biostatistics, Department of Global Health, Stellenbosch University, Francie Van Zijl Drive, Tygerberg, Cape Town 7505, South Africa; 4Western Cape Provincial Department of Health, 8 Riebeeck Street, Cape Town 8001, South Africa; hassan.mahomed@westerncape.gov.za; 5Division of Health Systems and Public Health, Department of Global Health, Stellenbosch University, Francie Van Zijl Drive, Tygerberg, Cape Town 7505, South Africa

**Keywords:** missed opportunities for vaccination, immunisation coverage, COVID-19, implementation research, quality improvement, South Africa

## Abstract

Despite South Africa’s substantial investments in and efforts at ensuring universal access to immunisation services, progress has stalled and remains suboptimal across provinces and districts. An additional challenge is posed by the ongoing coronavirus disease 2019 (COVID-19) pandemic, which has disrupted immunisation services globally, including in South Africa. While there is growing evidence that missed opportunities for vaccination (MOV) are a major contributor to suboptimal immunisation progress globally, not much is known about the burden and determinants of MOV in the South African context. Herein, we make a case for assessing MOV as a strategy to address current immunisation coverage gaps while mitigating the adverse impacts of the COVID-19 pandemic on immunisation services. We illustrate a practical implementation research approach to assessing the burden of MOV among children in primary care settings; for understanding the factors associated with MOV; and for designing, implementing, and evaluating context-appropriate quality improvement interventions for addressing missed opportunities. Such efforts are vital for building health system resilience and maintaining immunisation programme capacity to optimally deliver essential health services such as routine childhood immunisation, even during pandemics.

## 1. Introduction

Immunisation has long been recognised as one of the greatest advances in public health [1,2]. It is considered a highly cost-effective means of disease prevention, yielding significant cost savings for governments and individuals through averting ill-health, mortality, and long-term disability [3]. The immense public health and socioeconomic benefits of immunisation prompted the World Health Organization (WHO) to launch the Expanded Programme on Immunisation (EPI) in 1974, as the first ever global health effort to promote and ensure universal access to vaccines for all children [4,5]. This improved access to immunisation at the population level has helped to avert approximately four million child deaths every year from vaccine-preventable diseases [6,7,8]. These are in addition to the enormous societal and economic benefits of vaccination through the aversion of catastrophic health expenditures and the loss of human productivity due to ill-health [9].

Despite the substantial progress made in immunisation coverage since the inception of the EPI, coverage has stalled and remains suboptimal in many countries [10]. Global immunisation coverage has stagnated at 84–85% for a decade, far below the Global Vaccine Action Plan (GVAP) 2020 target of 90% coverage [10,11]. An estimated 20 million infants do not have access to vaccines or do not complete the vaccination series, the majority of whom reside in a few countries of sub-Saharan Africa, including South Africa [12,13,14]. These gaps are further complicated by the ongoing coronavirus disease 2019 (COVID-19) pandemic, which has disrupted essential health services nationally, including routine immunisation [15,16]. The disruption has resulted from a shift in the focus of health services to COVID-19 pandemic response, movement restrictions and physical distancing, and the fear of contracting COVID-19 during health facility visits [17].

## 2. Routine Childhood Immunisation in South Africa

Routine immunisation services in South Africa are provided at no cost through the Expanded Programme on Immunisation of South Africa (EPI-SA) [18]. The National Department of Health (NDOH) formulates policies, procedures, and guidelines to support routine immunisation service implementation across the country through the EPI-SA [18,19]. In May 2012, South Africa joined other WHO Member States to endorse the GVAP [11]. The GVAP envisioned a world in which all individuals and communities enjoy lives free from vaccine-preventable diseases, with a global target for countries to reach 90% national coverage of all their primary series vaccines by 2020 [10,11].

Like many other low- and middle-income countries, South Africa is experiencing enormous challenges with optimising its national immunisation coverage. Moreover, socioeconomic inequalities remain a huge problem in the post-Apartheid era, manifesting in disparities in health-related outcomes, including immunisation coverage [20]. While there have been significant investments to ensure universal access to immunisation services in South Africa, the country’s immunisation coverage is currently in stagnation, with data showing a steady decline since 2014 [21]. Data from the WHO and the United Nations Children’s Fund (UNICEF) estimates of immunisation coverage show a worrisome decline in coverage of the third dose of the diphtheria–tetanus–pertussis-containing vaccine (DTP3), from a peak of 85% of age-eligible children in 2014 to approximately 74% currently [21]. Moreover, our recent analysis of the 2016 South African Demographic and Health Survey data showed that as much as two-fifths (40.8%) of the country’s children are not fully immunised for age [22].

Hence, the value that vaccines deliver today in South Africa remains far below the substantial benefits they can offer [5]. The 2009–2010 measles outbreak and the more recent outbreaks in some provinces, including the 2017 measles outbreak in the Western Cape, have raised concerns about the country’s suboptimal levels of immunisation coverage [23,24]. Worse still, these gaps are likely to have been exacerbated by the current pandemic. Suboptimal immunisation coverage among children in South Africa has been attributed to several factors. These include a lack of awareness of the immunisation schedule by parents, non-attendance of antenatal care during pregnancy, and health workers’ inadequate immunisation training and heavy workloads [19,23,25]. Other factors include vaccine stock-outs, poor communication among stakeholders (including insufficient advocacy and inadequate social mobilisation), and weak collaboration between the public and private health sectors [5,26].

## 3. Missed Opportunities for Vaccination as a Driver of Suboptimal Immunisation Coverage

The WHO has recognised that missed opportunities for vaccination (MOV) are a major contributor to sub-optimal immunisation coverage. To accelerate and sustain immunisation coverage progress, the WHO recommends the provision of immunisation services at every contact with the health system [27]. A missed opportunity for vaccination refers to any contact with health services by an individual who is eligible for vaccination (unvaccinated or not up-to-date, and free of contraindications to vaccination) that does not result in the individual receiving all of the vaccine doses for which s/he is eligible [13]. MOV can occur in any of the following settings:During visits to health facilities (clinics and hospitals) or mobile health services for immunisation services;During visits to health facilities or mobile health services for other preventive services (e.g., growth monitoring sessions);During visits to health facilities or mobile health services for curative services (e.g., treatment of fever, cough, diarrhoea, or injuries);While accompanying a family member to a health facility for any type of service [13].

Common causes of MOV include the failure or inability of health providers to screen patients for eligibility, perceived contraindications to vaccination on the part of providers and parents, vaccine shortages, and rigid clinic schedules that separate curative services from vaccination areas [13].

The findings from our preliminary analysis of the 2016 South African Demographic and Health Survey data suggest that MOV constitutes a major burden, with a national prevalence of 42% and a Western Cape provincial prevalence of 36% [28]. The study identified individual-level factors, such as maternal attendance of antenatal care during pregnancy, and geographical factors, such as province of residence, as significant determinants of MOV.

## 4. COVID-19 and the Growing Need for Assessing and Addressing MOV

Since 1983, the WHO has recommended the provision of immunisation services at every contact with the health system as a strategy for improving immunisation coverage [27]. There has been an increasing momentum at global, national, and sub-national levels to conduct MOV assessments, to better understand the burden and its structural and contextual mechanisms [29,30,31]. In 2016, the Strategic Advisory Group of Experts (SAGE) on immunisation recommended the prioritisation of MOV assessments, based on evidence indicating a global MOV prevalence of 32% [31,32]. While there is currently limited evidence on the magnitude of MOV and the multi-dimensional nature of the factors associated with it, research efforts in this area have increased over the last decade, including in African and other low- and middle-income countries [29,30,33,34,35,36].

Although there is growing evidence that missed immunisation opportunities constitute a major burden globally and across sub-Saharan Africa [33,37,38,39,40], not much is known about the burden in South Africa. The various factors associated with the problem are less certain, as are the role of implementation research and the effectiveness of quality improvement interventions for addressing the problem. Although a few studies have assessed the burden of MOV in the South African context [23,41], none has implemented or evaluated the effectiveness of any intervention to address the burden.

The emergence of the COVID-19 pandemic and its far-reaching disruption of health services have exacerbated the gaping wounds of South Africa’s suboptimal immunisation coverage progress [17]. The disruption has been due to factors such as the national lockdown restrictions, the avoidance of health facilities due to the fear of contracting COVID-19, and the prioritisation of COVID-19-related health services, including the country’s current efforts to scale up COVID-19 vaccination coverage [42,43]. While the government has recommended that immunisation services continue uninterrupted during the lockdown period, there are indications that the pandemic has had negative impacts on essential health services, including routine childhood immunisation [16,42,43]. Evidence suggests that immunisation services may be more sensitive to movement restrictions in response to the pandemic, with child immunisation visits dropping by over 50%, while visits for most other services dropped less remarkably during the national lockdown period [42,43]. In addition, current anti-vaccine sentiments toward COVID-19 vaccines may likely have a negative impact on the routine immunisation programme, but this requires further investigation. The pandemic thus poses an additional threat to the country’s shaky progress with immunisation coverage.

To ensure health system resilience, there is a need to maintain the system’s capacity to deliver essential health services such as immunisation, even during pandemics or other crisis situations. As such, understanding the burden and determinants of missed opportunities for childhood vaccination and instituting remedial actions have become more imperative for mitigating the further disruption of immunisation services by the COVID-19 pandemic and fostering health system resilience in crisis settings [15]. Every health facility visit is a golden opportunity for catch-up vaccination of children that might have missed age-eligible vaccine doses during the lockdown period or prior to that. Efforts aiming to assess and address MOV will be in line with current national primary healthcare and immunisation service improvement mandates [44,45]. In addition to helping io identify actionable and sustainable change ideas that can be scaled up nationally and across provinces and districts for strengthening immunisation service delivery, such efforts can facilitate the attainment of the post-2020 GVAP targets and the immunisation-related targets of the Sustainable Development Goals (SDGs) [46,47].

## 5. An Implementation Research Approach to Assessing and Remedying MOV: Conceptual and Methodological Considerations

With the increasing momentum of efforts towards accelerating the attainment of universal health coverage (UHC), countries are often faced with difficult choices regarding the most effective use of available health resources, particularly in contexts of resource limitation, as well as competing healthcare needs and political priorities [48]. This has been exacerbated by the negative impact of the pandemic on country economies. Implementation research has emerged in response to the critical need for maximising implementation outcomes [49]. Defined as the “scientific study of methods to promote the adoption and integration of evidence based practices, interventions and policies into routine health care and public health settings,” implementation research seeks to use locally-derived research evidence to improve implementation outcomes [50].

Implementation research can be applied at various stages of implementation—before, during, and after implementation. At the pre-implementation stage, implementation research can be used to assess organisational readiness, identify potential implementation barriers or facilitators, inform planning and resource mobilisation, set implementation goals and targets, and tailor implementation strategies to meet set goals. At the implementation stage, it can be used to monitor implementation progress, track the utilisation of resources, and identify immediate implementation gaps. At the post-implementation phase, it can be used to evaluate what worked (implementation effectiveness and facilitators) and what did not work (implementation constraints and barriers), while enabling the use of such findings in strengthening implementation processes and fostering sustainability.

In the context of MOV, an implementation research approach to assessing and remedying the problem should integrate qualitative and quantitative research methods that are appropriate for assessing the magnitude and multi-dimensional determinants of MOV, while identifying, implementing, and evaluating the effectiveness of remedial measures. In accordance with the WHO’s MOV assessment strategy, it should aim to answer three important questions: (1) How many opportunities for vaccination are being missed?; (2) why are these opportunities being missed?; (3) what can be done to reduce missed opportunities? [13].

Such an approach should be underpinned by an understanding of the relationship between individual, contextual, and structural factors and MOV; how these are amenable to remedial measures through a theory of change; and how remedial measures can be evaluated and sustained using theory-informed evaluation designs. It should involve distinct but complementary mixed-methods research components that triangulate quantitative and qualitative data from a broad range of sources, including caregivers, health workers, and health facility managers. The following paragraphs describe what these various components might look like.

Figure 1 below illustrates an implementation research work plan for MOV assessment and remediation. This was adapted from the WHO’s MOV assessment methodology [13] and from previous implementation research approaches to improving health service delivery within African contexts [49].

### 5.1. MOV Assessment

This includes a facility-level quantitative assessment of the magnitude of MOV and the individual and contextual factors associated with it, complemented by a qualitative exploration of the factors associated with MOV from the perceptions of caregivers, health workers, and immunisation programme managers. In situations where facilities, districts or provinces have already documented evidence of the existence of MOV, it may not be necessary to conduct an MOV assessment. In such circumstances, facilities or immunisation programmes at the district and provincial levels may choose to move directly to implementing locally tailored remedial actions, as highlighted below.

### 5.2. Designing, Implementing, and Evaluating Remedial Actions against MOV

This can involve a quality improvement (QI) intervention designed and implemented based on qualitative and quantitative assessment findings. QI approaches have been increasingly used in healthcare to improve health outcomes over the years [51,52,53,54]. The WHO defines QI as “a change in process in a health-care system, service, or supplier to increase the likelihood of optimal clinical quality of care measured by positive health outcomes for individuals and populations” [55]. QI typically relies on the collaborative efforts of stakeholders to create and implement change ideas that can result in the desired outcomes [53]. The QI action plans should be adopted and implemented through an iterative process of collective dialogue, goal sharing, and collaborative problem sharing by a QI team—composed of immunisation programme managers, health facility managers, frontline health workers, and caregivers. It should also involve the continuous monitoring of MOV outcomes, in line with key performance indicators and targets agreed upon by the QI team.

### 5.3. Post-Implementation Evaluation

It is also necessary to conduct a post-implementation evaluation of remedial interventions implemented to address MOV. This is a major entry point for using theory-informed implementation research methods for assessing implementation outcomes while identifying individual and contextual factors influencing these outcomes, including the facilitators and barriers to implementation sustainability. The use of implementation research methods can aid the uptake, implementation, and translation of evaluation findings into routine implementation practices. Applicable implementation research methods in this context include the Consolidated Framework for Implementation Research (CFIR), Reach Effectiveness Adoption Implementation Maintenance (RE-AIM), and Theoretical Domains Framework (TDF) [56,57,58]. The implementation lessons learnt here are particularly useful for understanding what worked, what did not work, and how these can be contextualised for guiding and sustaining further MOV remedial efforts and implementation processes.

Overall, the pragmatic use of an implementation research approach can help to determine the burden of MOV and its underlying factors with contextual precision, while helping to understand the roles of structural agents and contextual factors in shaping the impact of remedial measures implemented for addressing MOV. In addition, through the participation of caregivers, health workers and immunisation programme managers in identifying and solving problems, such an approach allows key immunisation stakeholders to take ownership and responsibility for reducing missed opportunities. The real-world setting of the implementation research design also creates an opportunity for sharing implementation lessons for improving routine immunisation coverage at the facility, provincial, and national levels.

### 5.4. Resource Implications

Assessing and addressing the problem posed by MOV will require resources, such as the availability of vaccines and adequate supply of other immunisation service commodities and the needed logistic resources. It will also require resources for health service providers’ training; interdisciplinary collaboration between policymakers, immunisation programme managers, frontline implementers, and researchers, and adequate communication among these actors. To ensure the highest impact on reducing MOV, high-impact interventions that have the highest probability for implementation should be prioritised. These may include interventions that are relatively easy and low cost to implement, those that had been planned previously but are yet to be implemented (e.g., due to training gaps), or ongoing ones that require enhancement.

It is important that the plans are well incorporated into EPI programmatic workplans and facility-level patient flow to ensure sustainability. It is also critical that proposed MOV remedial action plans are assigned to specific persons or teams responsible for the immunisation programme for easy coordination, tied to an implementation timeline and targeted deliverables. To ensure funding and sustainability, it is vital that MOV assessment activities and remedial action plans be endorsed by key immunisation programme decision-makers and stakeholders, such as EPI programme managers, immunisation partners, and other relevant stakeholders at the local and national levels.

## 6. Conclusions

The emergence of the COVID-19 pandemic has exacerbated the gaping wounds of South Africa’s suboptimal progress in improving immunisation coverage. Available evidence suggests that missed opportunities for vaccination are a major driver of low immunisation uptake in South Africa. We made a case for an implementation research approach for assessing and remedying the burden of, and the factors associated with, missed opportunities for vaccination among children in primary care settings, applicable in the South African and similar contexts. Efforts in this direction will be important for building health system resilience and the capacity of immunisation programmes to optimally deliver vaccination services, even during pandemics.

## Figures and Tables

**Figure 1 vaccines-09-00691-f001:**
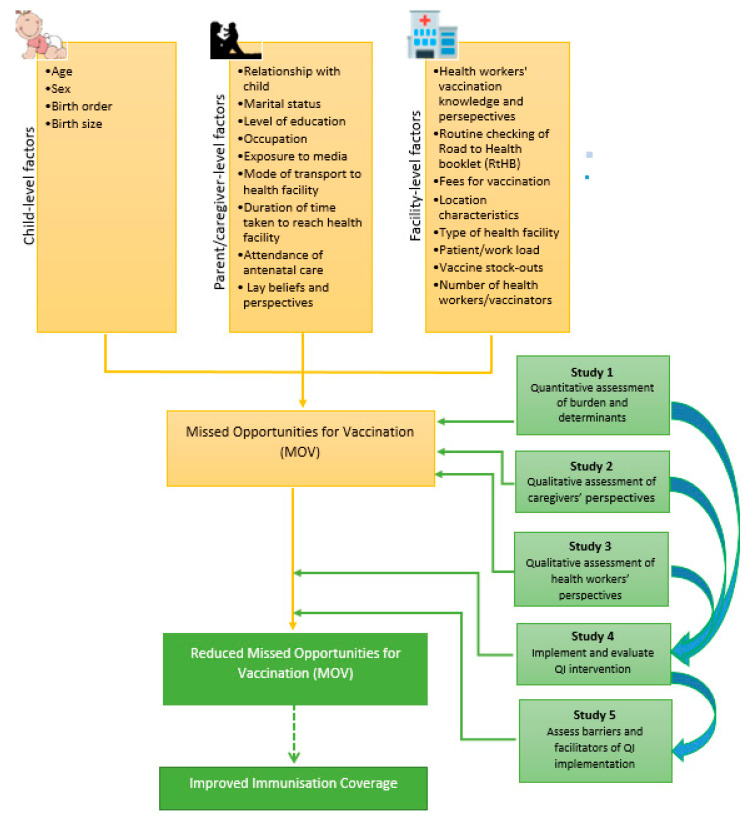
An implementation research work plan for MOV assessment and remediation.

## Data Availability

Not applicable.

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
