# Peer review of "COVID-19 and the Gaping Wounds of South Africa’s Suboptimal Immunisation Coverage: An Implementation Research Imperative for Assessing and Addressing Missed Opportunities for Vaccination"

_vaccines, 2021, doi:10.3390/vaccines9070691_

Round 1
Reviewer 1 Report
I do not have much to comment on, the literature is reviewed well and the manuscript flows nicely. I think the sections prior to COVID-19 could be kept briefer
more brief:
Strengths: The manuscript does a decent job capturing the progress made so far with immunizations and highlights the importance of addressing MOVs to further improve coverage. The authors have also explained MOVs well.
Suggestions:
- On certain occasions, it is not clear which year is being referred to, e.g. "...from a peak of 85% in 2014 to about 74% 79 currently.[21]" (page 2; line 80). Similarly, in "Furthermore, our recent analysis of a nationally-representative dataset 80 showed that as much as two-fifths (40.8%) of the country’s children are not fully immunised for age." (lines 80–82, page 2), which year was the data collected? Please specify these
- It will be helpful to contextualize if we know the specific year(s) being referred to by "more recent" (line 84, page 2)
- As the main focus is on South Africa, I do not think the general section on the global scenario adds much value.
- Given that COVID-19 is fundamental to the theme of the paper, what about the situation with the COVID-19 vaccines and how that may have strained the system even further? Is there evidence to suggest that routine immunization is getting lower priority since the advent of the vaccines and the scrambling to secure a supply of safe and effective COVID-19 vaccines? Or of manufacturing efforts being directed towards that end?
- Any precedence of similar implementation research approach elsewhere should be noted, say, if the plan is adapted from somewhere
Author Response
Comment: I do not have much to comment on, the literature is reviewed well and the manuscript flows nicely. I think the sections prior to COVID-19 could be kept briefer.
Response: We appreciate the kind words and have also shortened the background sections as suggested.
Comment: Strengths: The manuscript does a decent job capturing the progress made so far with immunizations and highlights the importance of addressing MOVs to further improve coverage. The authors have also explained MOVs well.
Response: Thank you for the commendation.
Comment: On certain occasions, it is not clear which year is being referred to, e.g. "...from a peak of 85% in 2014 to about 74% 79 currently. [21]" (page 2; line 80). Similarly, in "Furthermore, our recent analysis of a nationally-representative dataset 80 showed that as much as two-fifths (40.8%) of the country’s children are not fully immunised for age." (lines 80–82, page 2), which year was the data collected? Please specify these.
Response: These have been clarified (Lines 74-77)
Comment: It will be helpful to contextualize if we know the specific year(s) being referred to by "more recent" (line 84, page 2)
Response: This has been addressed (Line 80)
Comment: As the main focus is on South Africa, I do not think the general section on the global scenario adds much value.
Response: We appreciate this suggestion. Given that MOV is a global agenda that has created the impetus to assess and remedy it at national and sub-national levels, a review of the literature needs to account for the global landscape. Nonetheless, we have shortened and revised the background section to make it more nationally focused.
Comment: Given that COVID-19 is fundamental to the theme of the paper, what about the situation with the COVID-19 vaccines and how that may have strained the system even further? Is there evidence to suggest that routine immunization is getting lower priority since the advent of the vaccines and the scrambling to secure a supply of safe and effective COVID-19 vaccines? Or of manufacturing efforts being directed towards that end?
Response: There is currently no published evidence on the direct impact of the prioritization of COVID-19 vaccines on routine immunization. We have rather noted and provided local evidence on how the pandemic has generally affected health services, including routine immunization. (Lines 136 – 151)
Comment: Any precedence of similar implementation research approach elsewhere should be noted, say, if the plan is adapted from somewhere
Response: This has been added (Lines 206 – 208)
Reviewer 2 Report
This is a terrific opinion piece with sound methods recommended to improve immunization rates in South Africa. As a teacher of public health, I would use such a piece for classroom or school required field work, to teach students how to do public health quality improvement.
This reviewer did get a little confused with the many acronyms, and had to keep looking them up (e.g. LMIC). It might be reasonable to use fewer.
Line 62: by "freely" do you mean, at no cost to the patient?
Author Response
Comment: This is a terrific opinion piece with sound methods recommended to improve immunization rates in South Africa. As a teacher of public health, I would use such a piece for classroom or school required field work, to teach students how to do public health quality improvement.
Response: We very much appreciate this acknowledgement of the public health value of our work. We are glad to hear that you found it valuable enough to be used as a teaching resource.
Comment: This reviewer did get a little confused with the many acronyms, and had to keep looking them up (e.g. LMIC). It might be reasonable to use fewer.
Response: We have defined the acronym in full now (line 128).
Comment: Line 62: by "freely" do you mean, at no cost to the patient?
Response: That has been clarified (Line 57). Thank you.
Reviewer 3 Report
In the manuscript, the authors analyze the suboptimal immunization coverages which characterize South Africa. Childhood immunization, in particular, has been worsened by the COVID-19 pandemic. The recent data highlight that missed opportunities for vaccination (MOV) represent a major contributor to suboptimal immunization progress. The authors propose a model for the evaluation of the burden of MOV on immunization of children and suggest a model for improvement interventions for addressing missed opportunities.
The manuscript is well written and the topic is very interesting, representing an important key point in public health, in particular in the light of the change caused by the spread of SARS-CoV-2.
I have some minor comments.
- Lines 177-185: this paragraph should be rephrased since in this form it appears unclear.
- Line 319 (reference n.15): check the style of bibliographic reference.
- It would be interesting to add some information on anti-COVID19 vaccines in South Africa
Author Response
Comment: In the manuscript, the authors analyze the suboptimal immunization coverages which characterize South Africa. Childhood immunization, in particular, has been worsened by the COVID-19 pandemic. The recent data highlight that missed opportunities for vaccination (MOV) represent a major contributor to suboptimal immunization progress. The authors propose a model for the evaluation of the burden of MOV on immunization of children and suggest a model for improvement interventions for addressing missed opportunities.
Response: We appreciate these kind words.
Comment: The manuscript is well written and the topic is very interesting, representing an important key point in public health, in particular in the light of the change caused by the spread of SARS-CoV-2.
Response: We are grateful for this commendation of our work.
Comment: Lines 177-185: this paragraph should be rephrased since in this form it appears unclear.
Response: This paragraph has been revised for clarity (Lines 178 – 187)
Comment: Line 319 (reference n.15): check the style of bibliographic reference.
Response: Thank you for the observation. That has been rectified (Lines 336 – 340)
Comment: It would be interesting to add some information on anti-COVID19 vaccines in South Africa
Response: This has mentioned (Lines 138 – 141 and Lines 148 - 150).